# Synthesis of Fe_3_O_4_@mZrO_2_-Re (Re = Y/La/Ce) by Using Uniform Design, Surface Response Methodology, and Orthogonal Design & Its Application for As^3+^ and As^5+^ Removal

**DOI:** 10.3390/nano11092177

**Published:** 2021-08-25

**Authors:** Easar Alam, Qiyan Feng, Hong Yang, Jiaxi Fan, Sameena Mumtaz, Farida Begum

**Affiliations:** 1Engineering Research Center of Ministry of Education for Mine Ecological Restoration, China University of Mining and Technology, Xuzhou 221116, China; easaralam@hotmail.com (E.A.); caroline_yanghong@hotmail.com (H.Y.); fjx1239153497@163.com (J.F.); 2Department of Biological Sciences, Karakoram International University, Gilgit 15100, Pakistan; samina@kiu.edu.pk; 3Department of Environmental Science, Karakoram International University, Gilgit 15100, Pakistan; farida.shams@kiu.edu.pk

**Keywords:** magnetic Fe_3_O_4_, uniform design, surface response methodology, multi-staged doping, arsenic (III & V) removal

## Abstract

In this study, iron oxide (Fe_3_O_4_) was coated with ZrO_2_, and doped with three rare earth elements((Y/La/Ce), and a multi-staged rare earth doped zirconia adsorbent was prepared by using uniform design U_14_, Response Surface methodology, and orthogonal design, to remove As^3+^ and As^5+^ from the aqueous solution. Based on the results of TEM, EDS, XRD, FTIR, and N_2_-adsorption desorption test, the best molar ratio of Fe_3_O_4_:TMAOH:Zirconium butoxide:Y:La:Ce was selected as 1:12:11:1:0.02:0.08. The specific surface area and porosity was 263 m^2^/g, and 0.156 cm^3^/g, respectively. The isothermal curves and fitting equation parameters show that Langmuir model, and Redlich Peterson model fitted well. As per calculations of the Langmuir model, the highest adsorption capacities for As^3+^ and As^5+^ ions were recorded as 68.33 mg/g, 84.23 mg/g, respectively. The fitting curves and equations of the kinetic models favors the quasi second order kinetic model. Material regeneration was very effective, and even in the last cycle the regeneration capacities of both As^3+^ and As^5+^ were 75.15%, and 77.59%, respectively. Adsorption and regeneration results suggest that adsorbent has easy synthesis method, and reusable, so it can be used as a potential adsorbent for the removal of arsenic from aqueous solution.

## 1. Introduction

Arsenic is the 20th most toxic and carcinogenic metallic element found in nature, and its content in the earth crust is about 2–5 mg/kg [1]. People are exposed to arsenic mainly by the direct contact with drinking water and indirect contact through food chain transmissions. When arsenic and its compounds are ingested by the human body, they can accumulate in human body, and the main parts of human body that are prone to arsenic accumulation are hair, nails, bones, liver, kidney, etc. [2]. If the human body is directly exposed to a large amount of arsenic, the central nervous system transmission will be diminished, resulting in numbness of hands and feet, gastrointestinal and respiratory tract lesions etc. [3]. It is proven that arsenic is usually found in the form of arsenic oxides and arsenic trioxides. Inorganic arsenic is divided into As(III) and As(V). The toxicity level of arsenic(III) is higher than arsenic(V) [4]. Therefore, it is crucial to control arsenic pollution in the environment, especially in the water environment.

The most common methods used for arsenic removal include; precipitation, coagulation and sedimentation, electro-dialysis, electro-coagulation, reverse osmosis, nanofiltration, ion-exchange, oxidation, adsorption, etc. [3]. These technologies have their advantages, disadvantages, and conditions of use. There are also differences in the operating costs, simple usage, and long-term operational reliability [5,6]. Among these methods, adsorption is one of the most promising approaches for the treatment of arsenic polluted wastewater [7]. In the last many years, different potential adsorbents like hydroxides, activated carbon, graphene, activated alumina etc. have been synthesized, and used for arsenic removal [8]. Zirconia is one of the adsorbents used for arsenic removal and many studies suggests that Zirconia has a good adsorption effect for arsenic removal from the aqueous solutions. The ZrO_2_ layer can rise the adsorption properties of adsorbents to achieve the efficient removal of As(III) and As(V) from the water [9]. Rare earth doped nanomaterials have been synthesized and extensively used for water pollution, disease diagnosis, drug delivery, biocompatibility, drug loading etc. [10].

Uniform design is a statistical tool used for examining the relationship between various experimental variables to one or more responses [11]. Uniform design is particularly well suited to multi-factors, and multi-level assessments, such as assessing nanomaterial preparation molar ratios [12,13]. Response surface methodology (RSM) can give a spontaneous graph to impulsively observe the optimization points, and spontaneously determine the optimization areas [14]. Uniform Design and RSM together are very effective in optimizing the synthesis ratios of magnetic nanomaterials [15,16]. The orthogonal experiment method is a type of design method used to investigate a variety of factors and levels. It performs tests by choosing a suitable number of representative test cases from a large set of test data [17]. Although there are many studies on adsorption of arsenic have been reported. However, there are very limited or no studies on material optimizations using RSM, Uniform design, and also orthogonal design and multi doping of rare earth metals.

In this paper by using uniform design U_14_*(14^5^) and response surface analysis (RSM), the synthesis ratio of zirconium oxide was determined; by using L9 orthogonal design table the doping ratio of rare earth elements was determined. Where, iron ixide (Fe_3_O_4_) was coated with ZrO_2_ and doped with three rare earth elements (Y/La/Ce), so to prepare rare earth doped zirconium adsorbent to treat As^3+^ and As^5+^ from water. Through different characterization analysis like XRD, FT-IR, N_2_ adsorption-desorption, EDS, VSM, and TEM, the morphology, elemental composition, pore size, specific surface area, and crystallinity were determined. Then adsorption efficiencies of As^3+^ and As^5+^ on to the prepared material was examined.

## 2. Materials and Methods

### 2.1. Experimental Materials

China Pharmaceutical Group Chemical Reagents Co., Ltd. (Shanghai, China) supplied FeCl_3_-6H_2_O of Analytically pure grade. Analytically Pure Sodium Acetate was received from Xilong Science Co., Ltd. (Shantou, China). China Pharmaceutical Group Chemical Reagents Co., Ltd. provided pure Analytical Grade Nitric Acid (HNO_3_), Sodium Arsenate (Na_3_AsO_4_-12H_2_O), and Polyvinylpyrrolidone K30 (PVP. Absolute Ethanol (CH_3_CH_2_OH) was received from Shanghai Zhongqin Chemical Reagent Co., Ltd. (Shanghai, China). Shanghai Aladdin Biochemical Technology Co., Ltd. (Shanghai, China) provided Industrial grade Zirconium (IV) butoxide (Zr(OBu)_4_). Guangdong Weng Jiang Chemical Reagent Co., Ltd. (Shaoguan, China) produced analytically pure lanthanum nitrate hexahydrate La(NO_3_)_3_·6H_2_O, and Cerium nitrate hexahydrate Ce(NO_3_)_3_·6H_2_O. Analytically Pure Sodium Arsenite (NaAsO_2_) was purchased from Chemical Reagent Research Center of Guangdong (Guangzhou, China).

### 2.2. Characterization Methods

The micro-morphology of the sample was observed and examined using an American FEI Tecnai G2 F20 Field Transmission Electron Microscope (TEM), at 200 kV acceleration voltage. VSM was conducted by using a Vibrating Sample Magnetometer (Quantum Design, PPMS-9, San Diego, CA, USA) at a room temperature under the temperature-controlled conditions in the range of 2–400 K. A qualitative and quantitative study of the surface elemental composition of samples was carried out using an FEI Quanta FEG 250 Scanning Electron Microscope from the United States in the range of 200 V–30 kV electron beam voltage. FT-IR Spectrometer (VERTEX 80V, Brunker, Germany) was used for the characterization of groups on material surface. The sample was dried and dispersed in KBr powder and passed into pellets. Spectrum range: 4000–400 cm^−1^; resolution: 0.06 cm^−1^; beam diameter 40 mm. The The X-ray powder diffraction (XRD) patterns were collected by using X’PERT powder small angle X-ray diffractometer (PANalytical, Almelo, The Netherlands). The X-ray radiation source was a ceramic X-ray Diffraction Copper anode (Cu Kα radiation: λ = 1.54056 Å) and Bragg-Brentano configuration, and diffractions were taken at room temperature.

### 2.3. Synthesis of Fe_3_O_4_ and Fe_3_O_4_@mX-ZrO_2_, X = Y/La/Ce

Iron oxide (Fe_3_O_4_) was prepared using our previously reported work. 80 mL Ethylene Glycol, 4.50 g FeCl_3_-6H_2_O and 10g CH_3_COONa were first added in a round bottom flask, and mixed it well. 3.2 g TMAOH was separately dissolved in 20 mL Ethylene glycol and then added dropwise to the main solution, which was then mixed for 30 min using magnetic shaking. The solution was then transferred to a Teflon-lined stainless steel Autoclave and heated at 200 °C for 8 h. After cooling down at room temperature, the prepared MNPs were then cleaned several times with alcohol and deionized water, and dried in an oven at 50 °C for 12 h [16]. Then Fe_3_O_4_@mX-ZrO_2_, X = Y/La/Ce was prepared, where 1 g of PVP was first weighed in 20 mL of deionized water and after ultrasonic dispersion, a certain amount of magnetic Fe_3_O_4_ was added, and again the ultrasonic dispersion continued for 30 min. Then, the required amount of TMAOH was added, and continued ultrasonic shaking for 10 min. Meanwhile, 10mL ethanol, zirconium butoxide, rare earth elements (Y/La/Ce) were added into small glass vials and ultrasonically shook for 10 min. Then, these were added dropwise to the main solution of the system, and again ultrasonic shaking for 30 min, later solution was transferred to a three-mouth round bottom flask, and stimulated at room temperature for 3 h at 200 r/min. Subsequently, the solution was transferred to an oven and heated at 150 °C for 3 h for the hydrothermal reaction. Lastly, after cooling down, material was washed with alcohol and deionized water, and dried in the vacuum oven at 60 °C for 12 h.

### 2.4. Optimizing the Synthesis Ratio of Fe_3_O_4_@mZrO_2_-Re (Re = Y/La/Ce)

#### 2.4.1. Uniform Design

Uniform design is the statistical technique mostly used to study the relationship between different experimental factors to one or many related responses. In most of these cases, to run a complete factorial design to get sufficient resources may not always exist, so, therefore, small factorial designs are frequently used to significantly reduce the experiment numbers [11,12]. The uniform design table, the supporting tables, and the seven-level design are shown in Table 1, Table 2, Table 3 and Table 4. In Table 1 independent variables and experiment sequence is arranged and selected.

The rules for creating the columns of U_14_*(14^5^) are given in Table 2 and on the basis of this further designing of the selected factors was achieved.

By using the Table 1 and Table 2, seven levels of the three selected variables i.e., Fe_3_O_4_ dosage (X_1_), TMAOH dosage(X_2_), and zirconium butoxide dosage (X_3_) were derived, and laid a foundation for further use of uniform design. The seven levels of the selected variables are listed in Table 3.

The regression analysis model formula was used for analyzing the influencing factors of each variable, and to estimate, evaluate, and optimize the experiment conditions. The autoregressive-response model was fitted with quadratic polynomials since this model clearly expresses the interaction between independent variables. The fitting method is stepwise regression method (Equation (1)). Specific experimental arrangements and results are shown in Table 4.
(1)$$\;\;\;\;\;\;\;\;\;\;\;Y=\beta_{0}+\overset{m}{\underset{i=1}{\sum }}\beta_{i}X_{i}+\overset{m}{\underset{i=1}{\sum }}\beta_{ij}X_{i}{}^{2}+\overset{m}{\underset{i<j}{\sum }}\beta_{ij}X_{i}X_{j}+\varepsilon$$

Based on the designed U_14_ uniform experiment design, the results for each level was measured and identified the best combinations for adsorption. The amount of As^5+^ adsorbed on material surface at equilibrium was the response value; each factor was set at seven levels to investigate the relationship between factors and their impact on response values.

#### 2.4.2. RSM Analysis

RSM can determine the effects of various factors and their relations to the investigated indicators (response values) in scientific experiments. It can also be used to fit a comprehensive quadratic polynomial model through a fundamental extravagant experiment, which can give better experimental design and results [18]. For RSM, the main selected factors were; factor-1: Fe_3_O_4_, factor-2: TMAOH, and factor-3: Zirconium(IV) Butoxide, and the main factor chosen for this was Zirconium(IV) Butoxide.

#### 2.4.3. Orthogonal Design

To study the method of multi-factor and multi-level design, and to find the optimal level combination through a small number of experiments, orthogonal experiment design can be used. The larger the number and level of factors, the clearer the benefits of this method. Orthogonal experiment design is an efficient, fast, and economic experiment design method based on the orthogonal table [19]. To calculate the doping amount of the rare earth elements orthogonal design was applied. The arrangement of every factor and levels were optimized by using the orthogonal design of L9. The optimum ratio of the selected doping elements for the preparation of doped material were, A (Yttrium), B (Lanthanum), C (Cerium), and D (empty list), and the dosage of every cube of doping material in parentheses.

### 2.5. Adsorption Procedure

By using Sodium Arsenite (NaAsO_2_) and Sodium arsenate (Na_3_AsO_4_·12H_2_O) stock solutions of As^3+^, and As^5+^ were prepared and used for the adsorption experiments. The standard solutions were gradually diluted by using the stock solutions. The accurate weighing 0.01 g adsorbent was put into a 250 mL conical flask with a stopper and added 100 mL solutions of As^3+^, and As^5+^. The solution was adapted to the appropriate pH value by using HCL and NaOH. The tapered flask was then put in a persistent temperature oscillator at 200 r/min for 60 min. After one hour of oscillation, magnetic separation was used, and the supernatant was filtered by a membrane (0.45 μm), and concentration of the heavy metals was detected by ICP. The formula for adsorption capacity is as follows.
(2)$$Q_{e}=\frac{\left(C_{0}-C_{e}\right)\times \nu }{m}$$
where; *Q_e_* (mg/g) is the adsorption equilibrium capacity, *C*_0_ (mg/L) is the initial ion concentration, *C_e_* (mg/L) after adsorption ion concentration in solution, *V* (L) is the volume of the solution to be adsorbed, L; and *m* (g) is the adsorbent dosage.

## 3. Results and Discussions

### 3.1. Use Uniform Design, RSM and Orthogonal Design for Optimizing Adsorbent

#### 3.1.1. Uniform Design

On the basis of guidelines in the Table 1, Table 2 and Table 3 of the uniform design, the selected variables and their concentrations at designed 7 levels were calculated, and data is shown in the Table 4. X_1_, X_2_, and X_3_ were selected as the independent variables, which were Fe_3_O_4_ dosage, TMAOH dosage, and zirconium butoxide dosage, respectively. For each adsorption level, As(V) concentration was taken as 5 mg/L, volume 100 mL, adsorbent 0.01 gm, adsorbed at room temperature for 60 min, and performed the experiments at room temperature.

To get the adsorption capacity model of the prepared adsorbent for As^5+^ removal, the DPS-7.05 (Data Processing System) was used to choose the quadratic polynomial stepwise regression method to achieve regression analysis. The data processing system (DPS) is a comprehensive and user-friendly platform for experimental designs, statistical analysis, and data mining that provides uniform design and calculation benefits [20]. Equation (3) was used to examine the final values of Y‘, which are the predicted values of equilibrium adsorption capacities, and results are mentioned in Table 4.
(3)$$\begin{array}{c} Y‘=-2.762519818+179.02323398X_{3}+235.52925909X_{4}-5632.701746{X_{1}}^{2}-35731.654194{X_{3}}^{2}-\\ 4378.245969{X_{4}}^{2}-8720.569751X_{1}\times X_{2}+2660.3369935X_{1}\times X_{4}+12910.796840X_{2}\times X_{3} \end{array}$$

Fisher variance is a substantial tool to find the differences of means between two or more samples. In order to investigate the differences between the experimental and predicted values, Fisher’s variance test was used. If compared with given standard F-value, the larger the F-value is, the clearer the treatment result, and also test precision is higher [16]. Uniform design samples were prepared according to the range and level listed in the Table 4. The variance test of Experimental values (Y) and predicted values (Y‘) of the samples are shown in Figure 1.

With variance test, the Fisher variance test F-value is 13.38, which is much higher than F_(0.05,8,5)_ = 4.81. The correlation coefficient R^2^ is 0.9683, which is in good agreement with the experimental value Y, showing that only 3.17% total variance could not be explained by the model. Therefore, Equation (3) can be used to predict the molar ratio of the samples.

#### 3.1.2. RSM Analysis

The factors selected for RSM analysis were; Factor-1: Fe_3_O_4_, Factor-2: TMAOH, and Factor-3: Zirconium(IV) butoxide (Zr(OBu)_4_), and Zirconium(IV) butoxide were selected as the main factor. The Figure 2 and Figure 3 explains the model and methodology used to find the best adsorption combinations for As^5+^ removal from the aqueous solution.

From Figure 2 can be noticed that the response surface exhibits a clear frontal. The adsorption response surface of As^5+^ is curved with the change of Zr(OBu)_4_ dosage, which indicates that it is a vital factor to affect the adsorption process. With the increase in Zr(OBu)_4_ dosage, the adsorption also increased, but after sometimes it decreases, so an appropriate amount of Zr(OBu)_4_ dosage improved the adsorption [15]. From the contours of the interaction between Zr(OBu)_4_ and Fe_3_O_4_, it can be noticed that the high-value region appeared in the middle region i.e., Zr(OBu)_4_: 0.005~0.0058 mmol, and Fe_3_O_4_: 0.00048~0.00055 mmol.

From Figure 3 it can be noted that the adsorption response surface of As^5+^ is curved with the change of Zr(OBu)_4_ and TMAOH dosage, which indicates that both are crucial factors to affect the adsorption effect. From the contours of the interaction between TMAOH and Zr(OBu)_4_, it can be noticed that the high-value region appeared in the middle region (TMAOH: 0.0058~0.0065 mmol and Zr(OBu)_4_: 0.005~0.0058 mmol).

#### 3.1.3. Orthogonal Design

Based on the orthogonal design As(III) adsorption results the best doping combinations for adsorbent were finalized. The molar ratio of Fe_3_O_4_:TMAOH:Zr(OBu)_4_ for this design was kept at 1:12:11. The orthogonal design arrangement of each rare earth element (La/Ce/Y) and level is given in Table 5.

The revealed data from the Orthogonal design L9 (Table 5) suggests that the excellent combination was noted at level A2B1C2. Based on analyzed adsorption results, the optimal mixture ratio for the doping combination of the rare earth metals (Y/La/Ce) to prepare the best doping combinations for the removal of As(V) was finalized, which was as follows; Y: 1%; La: 0.2%; Ce: 0.8%. The relative molar ratio of Fe_3_O_4_:TMAOH:Zirconium(IV) butoxide was determined as 1:12:11, while the final best doping combination was found to be Y: 1%: La: 0.2%: Ce: 0.8. The best molar ratio for Fe_3_O_4_:TMAOH:Zr(OBu)_4_:Y:La:Ce was achieved as 1:12:11:1:0.02:0.08.

### 3.2. Characterizations of Fe_3_O_4_@mZrO_2_-Re (Re = Y/La/Ce)

#### 3.2.1. TEM Analysis

The morphology of material before and after surface modifications was tested by Transmission election microscopy (TEM). Figure 4 shows the TEM images of Fe_3_O_4_ (a), and Fe_3_O_4_@mZrO_2_-Re (Re = Y/La/Ce) (b). Uncoated Fe_3_O_4_ were observed as dispersed spherical particles with the diameter ranging 0.032–0.065 um. The good dispersibility probably originates from the TMAOH that effectively prevented from the agglomeration [21]. By contrast, the surface of Fe_3_O_4_@mZrO_2_-Re (Re = Y/La/Ce) exhibited uneven and larger morphologies probably due to the coating with ZrO_2_ and doped with rare earth metals.

#### 3.2.2. VSM Analysis

Magnetic saturation intensity is a key factor to check the magnetization of any magnetic material. The magnetic characteristics of the materials was investigated by Vibrant Sample Magnetometer (VSM) at room temperature. The magnetic hysteresis loops of the Fe_3_O_4_ and Fe_3_O_4_@mZrO_2_-Re (Re = Y/La/Ce) mesoporous materials are shown in Figure 5. As shown in the Figure 5 that the saturation magnetization values of Fe_3_O_4_ and Fe_3_O_4_@mZrO_2_-Re (Re = Y/La/Ce) are 49 emu/g and 19.8 emu/g, respectively. It can be observed that after the coating with ZrO_2_ and doping with rare earth metals, the magnetic saturation intensity significantly reduced i.e., from 49 emu/g to 19.8 emu/g. The decrease in the magnetic intensity saturation may be because of the addition of layers on the surface of the Fe_3_O_4_ but still, the magnetization value was enough for the magnetic separation in a short time by using an external magnet. The same kind of results are also described in other related studies [16,22].

#### 3.2.3. EDS Analysis

The Energy Dispersive Spectroscopy (EDS) of Fe_3_O_4_@mZrO_2_-Re (Re = Y/La/Ce) was conducted to determine the constituent elements, and corresponding peaks are shown in Figure 6, and elemental composition is listed in Table 6.

The EDS results in Figure 6 and Table 6 confirmed the presence of Fe, O, Zr, Ce, Y, and La, indicating that the prepared material have successfully coated with zirconia and doped with rare earth elements on the surface of Fe_3_O_4_. Table 6 shows that the weight% of Yttrium (Y), Cerium (Ce), and Lanthanum (La) is zero before doping, while it shows as 1.08%, 1.02%, and 0.45%, respectively, after doping, which confirms the doping.

#### 3.2.4. XRD Analysis

The XRD patterns of Fe_3_O_4_@mZrO_2_-Re (Re = Y/La/Ce) are shown in Figure 7. The XRD profile with weak signals towards the halo peak probably indicate that the magnetite contains both crystalline and amorphous parts [23]. The diffraction peaks at 2θ = 30.4, 35.8, 38.2, and 43, are assignable to the 220, 311, 200, and 400 Lattice planes of face centered-cubic magnetite (JCPDS No.19-0629), respectively [22].

#### 3.2.5. FTIR Analysis

FT-IR (Fourier transform infrared spectra) characterization was done to detect the major structural groups present in Fe_3_O_4_@mZrO_2_-Re (Re = Y/La/Ce). FTIR spectra of the material is shown in Figure 8.

As shown in Figure 8 that the FTIR spectrum of Fe_3_O_4_@mZrO_2_-Re (Re = Y/La/Ce) NPs stretches from 4000–400 cm^−1^, where the FTIR peaks at 498 cm^−1^ are likely attributed to Fe-O Stretching [24]. The characteristic band at 3419 cm^−1^ and 1623 cm^−1^ represent the stretching and bending of modes of OH^−^ group of the adsorbed water present on the surface of the nanomaterials [25]. The band appeared at 602 cm^−1^ of the spectra are assigned to the zirconia group. The peaks with obvious fluctuation are all related to hydroxyl groups. The valence band electrons of ZrO_2_ can be excited under certain conditions, and an electron-hole pair can be formed on its surface. The hole then can react with oxygen ion on the surface of ZrO_2_ to form an oxygen hole. Under the influence of the polar Zr-O bond, it dissociates to form a surface hydroxyl [26]. However, ZrO_2_ doped with Y, Ce, and La is likely to enhance the volume of hydroxyl groups at the surface of ZrO_2_ to upgrade its adsorption capacity.

#### 3.2.6. N_2_ Adsorption-Desorption Analysis

The porous characteristics of Fe_3_O_4_@mZrO_2_-Re (Re = Y/La/Ce) was examined by nitrogen adsorption isotherm measurement. The specific surface area and pore size distribution of the materials was measured from the analysis of desorption branch of isotherms using the density function theory.

As shown in Figure 9, an isotherm is typical for mesoporous material with a hysteresis loop at partial pressures [27]. According to Brunauer-Emmett-Teller (BET) analysis, the prepared materials exhibited large specific surface area of 265 m^2^/g, pore volume 0.156 cm^3^/g, and pore diameters 2–5 nm, respectively. N_2_ adsorption-desorption isotherms of the materials displayed conventional type IV curves with a sharp uptake at a high relative pressure, which demonstrates the existence of cavities between particles.

### 3.3. Adsorption Performance of Fe_3_O_4_@mZrO_2_-Re (Re = Y/La/Ce)

#### 3.3.1. Effect of pH

The initial pH value of the solution has a significant impact on metal ion adsorption. This parameter is directly related to the capability of hydrogen ions and metal ions to adsorb on the active surface sites [28]. Therefore, the effect of initial pH value on the adsorption performance of Fe_3_O_4_@mZrO_2_ (Re = Y/La/Ce) was investigated. 10 mg of adsorbent was accurately measured and put into different conical flasks, and added 100 mL of 5 mg/L solutions of As^3+^ and As^5+^, and adjusted the pH to 3.0, 4.0, 5.0, 6.0, 7.0, 8.0 and 9.0 by using HCL and NaOH. The remaining process was done by following the method in the Section 2.5.

From the Figure 10, it can be clearly noticed that the initial pH value of the solution has a substantial effect on the adsorption. Under the strong basic conditions, the adsorption capacity of the adsorbent for As^3+^ and As^5+^ removal was extremely low, while under weak acidic conditions, the adsorption capacities of both metals were high. The highest adsorption capacities for both As^3+^ and As^5+^ were recorded at pH 6 and pH 5, respectively. The adsorption capacity of As^3+^ from pH 5~pH 7 was almost near i.e., 30 mg/g, 32 mg/g and 30 mg/g, respectively. Based on initial pH studies the ideal adsorption pH conditions for both As^3+^ and As^5+^ were selected as 6 and 5, respectively.

The relevant studies suggest that that mesoporous zirconia-based nanostructures demonstrated good results for As(V) removal under lower acidic and neutral pH conditions [29]. For as (V), it mainly exists in the form of H_2_AsO_4_^−^ and in the range of pH = 3~7. At this time, the adsorbent surface has a positive charge, and the negatively charged H_2_AsO_4_^−^ will be adsorbed on the adsorbent surface directly through electrostatic attraction. With the increase of pH, the positive charge on the surface of the adsorbent decreases gradually, so the adsorption capacity of the adsorbent for As^5+^ decreases gradually; When the pH is greater than 7, the surface of the adsorbent begins to be occupied by negative charges [30]. Some other studies have shown that under acidic conditions, the surface of metal oxides prone to “protonation”, so the adsorption is negatively charged H_2_AsO_4_^−^ and HAsO_4_^2−^, and the protonation decreases with the increase of pH value. The protonation of the sorbent surface is supported by a lower pH. Improved protonation is supposed to escalate the positively charged spots, increase the attraction force between the sorbent surface and As anions, and hence boost adsorption in the lower pH range. The negatively charged sites dominate at higher pH levels, the repulsion effect increases, and the number of adsorption decreases. When the solution is alkaline, OH^−^ in the system will compete with arsenate anion for adsorption, resulting in the decrease of arsenic removal efficiency [31]. May be the same inclination was observed with the As^3+^, although the decline in As^3+^ removal was not that clear as that of As^5+^ with the increasing pH.

#### 3.3.2. Adsorption Isothermal Analysis

For non-linear fitting adsorption isothermal analysis, the three widely used isothermal adsorption models are the Langmuir, Freundlich, and Redlich–Peterson models [32]. The specific equation and application scope of each model are as follows:

Langmuir Isothermal non-linear adsorption equation is;
(4)$$q_{e}=\frac{q_{m}K_{L}C_{e}}{1+K_{L}C_{e}}$$
where: *q_e_* (mg/g) adsorption capacity at equilibrium, *C_e_* (mg/L) is equilibrium concentration, *b* (L/mg) is the Langmuir constant, and *q_m_* (mg/g) is the Saturated adsorption capacity.

*R_L_* is a dimensionless constant separation factor. Its definition formula is;
(5)$$R_{L}=\frac{1}{1+b\cdot C_{0}}$$

For the adsorption process on the heterogeneous surfaces, Freundlich isotherms are more suitable. The isotherm provides expressions for surface heterogeneity as well as the exponential distribution of active centers and their associated energies. Non-monolayer adsorption is the most common in this method [33]. The equation is as follows:(6)$$q_{e}=K_{F\;}\cdot {C_{e}}^{1/n}$$
where: *q_e_* (mg/g) is adsorption capacity, *K_F_* is the Freundlich coefficient (mg/g (L/mg)^1/*n*^), *C_e_* (mg/L) is Equilibrium concentration, and “*n*” is the Freundlich constant.

The Redlich–Peterson model combines the Langmuir and Freundlich models formula to summarize the Redlich–Peterson isothermal model [34], and the equation is as follows;
(7)$$q_{e}=\frac{K\cdot C_{e}}{1+\alpha \cdot {C_{e}}^{\beta }}$$where: *K*, α are Redlich–Peterson constant and “*β*” is coefficient.

The accurately weighted 10 mg adsorbent was put into different 250 mL conical flasks, and 100 mL solutions of As^3+^, and As^5+^ with different concentrations like 5 mg/L, 10 mg/L, 20 mg/L, 40 mg/L, 60 mg/L, and 80 mg/L were added, respectively. The solution was adjusted by HCL and NaOH to the required pH value. The conical flasks were then put in the constant temperature oscillator at 298 K, 308 K, and 318 K temperature conditions oscillated at 200 r/min for 60 min. After one hour of oscillation, magnetic separation was used, and the supernatant was then filtered through membrane (0.45 μm), and heavy metals concentration was detected by ICP-OES. According to change in the Ion concentration before and after adsorption the adsorption isotherm was drawn. Figure 11 and Table 7 shows the different adsorption isothermal curves and parameters of the models.

From the Isothermal fitting curves (Figure 11), and equations parameters of the Langmuir model, Freundlich model, and Redlich–Peterson Model, the best adsorption isothermal models for As^3+^ and As^5+^ removal was evaluated. The Langmuir model effectively described the adsorption isotherm data with all correlation coefficient (R^2^) ranged between 0.995~0.997 for As^3+^, and 0.988~0.990 for As^5+^. The R_L_ values calculated by the Langmuir model at different temperatures for As^3+^ and As^5+^ were ranged from 0.389~0.954, and 0.473~0.531, respectively, all were found less than 1, which suggested that the adsorption is in the favorable direction [35]. The results were found consistent, and the Qm fitted by the Langmuir model is slightly larger than the actual experimental results. The highest adsorption capacity of As^3+^ ions at 298 K, 308 K, and 318 K, were 62.00 mg/g, 64.31 mg/g, and 68.33 mg/g, respectively, and for As^5+^ at different temperatures were 59 mg/g, 64 g/g, and 84.23 mg/g, respectively. Based on R^2^ analysis, the Langmuir model and Redlich–Peterson model shows better results than the Freundlich model, indicating they are best fitted for As^3+^ and As^5+^ adsorption isotherm. Based on the analyzed isothermal results of the adsorption isotherms of As^3+^, and As^5+^ ions are mainly monolayer layer adsorption and absorption sites on the adsorbent are distributed homogenously [36].

The comparison of the prepared adsorbent Fe_3_O_4_@mZrO_2_-Re (Re = Y/La/Ce) with some relevant studies is shown in Table 8, where the comparison results revealed that the adsorbent is very effective in removing As^3+^ and As^5+^ from the aqueous solution.

#### 3.3.3. Adsorption Kinetics of Fe_3_O_4_@mZrO_2_-Re (Re = Y/La/Ce)

The two widely used kinetics models are; (1) the Pseudo-First order kinetic model, a basic kinetic rate equation of stable adsorption reaction; (2) the second one is Pseudo-second order reaction kinetics model, where adsorption rate of adsorbent is directly proportional to the number of available active sites on the surface [45,46,47]. In this study, pseudo-first-order and pseudo-second-order linear models were used to establish the rate constant and As(III) and As(V) controlling mechanism of adsorption onto the prepared adsorbent.
(1)Pseudo-first-order kinetics model
(8)$$\frac{dq_{t}}{dt}=k_{1}\left(q_{e}+q_{t}\right)$$

Converting the above equation into a linear equation:(9)$$\mathrm{Log}\left(q_{e}-q_{t}\right)=\log q_{e}-\frac{k_{1}}{2.303}t$$
where: *q_e_* (mg/g) is adsorption capacity at equilibrium, *q_t_* (mg/g) is the adsorption capacity of materials at time “*t*”, *t* (min) is adsorption reaction time, and *K*_1_ (min^−1^) is the first order adsorption rate constant.
(2)Pseudo-second-order kinetics model
(10)$$\frac{dq_{t}}{d_{t}\;}=K_{2}{\left(q_{e}-q_{t}\right)}^{2}$$

The equation given above is converted into a linear equation:(11)$$\frac{t}{q_{t}}=\frac{1}{K_{2}{q_{e}}^{2}}+\frac{1}{q_{e}}t$$
where: *q_t_* (mg/g) is the adsorption capacity at equilibrium, *q_t_* (mg/g) adsorption capacity at time “*t*”, *t* (min) is the adsorption reaction time, and *K*_1_ (min^−1^) is the second kinetic order adsorption rate constant.

Adsorption kinetic tests were conducted with 0.01 g adsorbent at initial concentrations of (1.0 mg/L, 2.0 mg/L, and 5.0 mg/L), pH 6.0, temperature 298 K, and at time durations of 10, 20, 40, 60, 80, 100, and 120 min. First and second linear kinetic models using Origin 8.5 software were applied to establish the rate constant and As^3+^ and As^5+^ adsorption controlling mechanism. The R^2^ (correlation coefficient) was used to find the most suited model. The values of K_1_, K_2_, and *q_e_* were also calculated.

From the fitting curves (Figure 12) and related parameters (Table 9) fitted by the kinetic models, it can be noticed that the R^2^ values obtained from the fittings of the two kinetic reaction models are significantly different. Among the two selected models, Pseudo-second-order kinetic model has a suitable effect, and R^2^ values were found for both As^3+^ and As^5+^ between 0.991–0.997, and found relatively stable, because there is no obvious fluctuation noted. The range of R^2^ values fitted by the Pseudo-first-order kinetics model are relatively wide ranged for both heavy metals, with values ranging from 0.921 to 0.987, with noticeable fluctuations. Therefore, the adsorption kinetics of As^3+^ and As^5+^ from solution by the adsorbent Fe_3_O_4_@mZrO_2_-Re (Re = Y/La/Ce) is more suitably designated by the Pseudo-second-order reaction kinetic model, where rate of reaction depended directly on the square of the concentration of the reactants remaining in the solution [48]. Many other researchers suggested that adsorption kinetics of Magnetic nanomaterials are supporting the Pseudo-second order Kinetics, and our results are also favoring the 2nd order kinetic model [49,50,51].

#### 3.3.4. Regeneration Efficiency of Fe_3_O_4_@mZrO_2_-Re (Re = Y/La/Ce)

Regeneration of materials is one the most important indicator to check the worth of any adsorbent, and the adsorbents with good regeneration abilities are very vital for the cost effective and environmental friendly adsorption process [52]. The regeneration experiment was performed at room temperature where 0.1 g of adsorbent was put in a 250 mL conical flask, and added 100 mL of As^3+^ and As^5+^ solutions with a concentration of 100 mg/L, and adjusted the pH to 5. Then placed the conical flask in a constant temperature oscillator for 3 h oscillated at 200 rpm. After filtration, determined the ion concentration in the filtrate to calculate the adsorption capacity. After every cycle NaOH solution was used as an elution agent, then washed with deionized water and dried in oven, and repeated the experiment for seven cycles. The regeneration experiment results of the adsorbent Fe_3_O_4_@mZrO_2_-Re (Re = Y/La/Ce) at different levels are shown in Figure 13.

In the first regeneration cycle, the regeneration efficiencies of both As^3+^ and As^5+^ were 93.45% and 95.75%, respectively. It can be seen that the regeneration performance of Fe_3_O_4_@mZrO_2_-Re (Re = Y/La/Ce) for As(III) and As(V) slightly decreases with the increase of regeneration cycles, but the adsorption efficiency of the adsorbent in all the later regeneration experiments tended to be stable. In the 7th cycle, the adsorption capacities of both As^3+^ and As^5+^ were 75.15% and 77.59%, respectively. The amount of adsorption slightly decreased after every cycle, and the possible reason for this loss is the structural uncertainty of the adsorbents, which during regeneration treatment might lose their adsorption sites and the structural order accessibility in the pores [53]. By regenerating the nanomaterials during wastewater treatment, we can make the removal process more viable and economical.

## 4. Conclusions

The optimum synthesis and doping ratios of the Fe_3_O_4_@mZrO_2_ (Re = Y/La/Ce) were determined by uniform design, surface response methodology, and L9 orthogonal design. Based on TEM, SEM, N_2_ adsorption desorption test, VSM, and XRD results, the suitable molar ratio of Fe_3_O_4_:TMAOH:Zirconium Oxide:Y:La:Ce was determined as 1:12:11:1:0.02:0.08. Prepared materials were characterized for morphology, crystallinity, magnetism and porosity etc. The prepared adsorbent materials were mostly spherical particles, and the outer layer of zirconium is mainly in the form of amorphous zirconia, which has good magnetic separation performance, specific surface area, and pore structure. The magnetic saturation capacity of Fe_3_O_4,_ and Fe_3_O_4_@mZrO_2_-Re (Re = Y/La/Ce) were 49 emu/g and 19.8 emu/g, respectively. The specific surface area was 263 m^2^/g, porosity was 0.156 cm^3^/g, and the surface has rich hydroxyl groups. The optimal initial pH values of the system were 6 (As^3+^) and 5 (As^3+^). The results of isothermal curves and fitting equation parameters shown that Langmuir model and Redlich Peterson model fitted well, while Freundlich model has a low fitting degree. As per calculations of the Langmuir model the highest adsorption capacities for As^3+^ and As^5+^ ions were recorded as 68.33 mg/g, 84.23 mg/g, respectively. From the fitting curves of the selected kinetic models, the adsorption process is more favors the quasi second order kinetic model, while first kinetic order was on lower side. The material regeneration was very effective, so it is suggested to be used as a potential adsorbent for the removal of arsenic-based pollutants from water.

## Figures and Tables

**Figure 1 nanomaterials-11-02177-f001:**
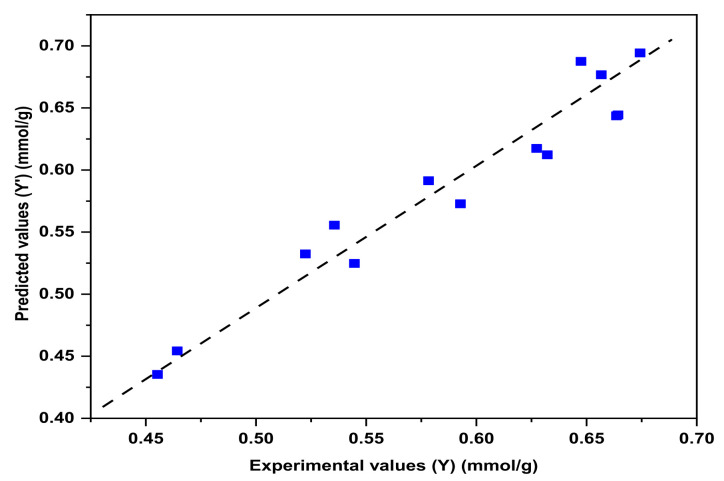
Relationship between Y and Y‘ values.

**Figure 2 nanomaterials-11-02177-f002:**
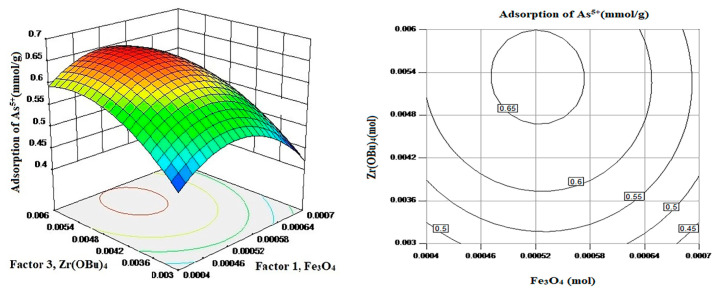
Response surface and contour of As^5+^ removal for Factor 1 & 3 interaction.

**Figure 3 nanomaterials-11-02177-f003:**
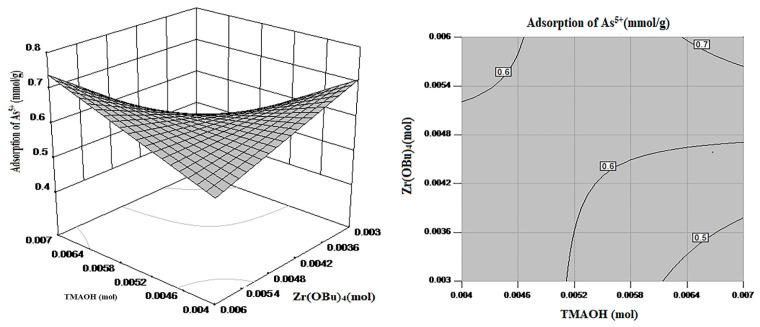
Response surface and contour of As^5+^ removal under the interaction of Factor 2 and 3.

**Figure 4 nanomaterials-11-02177-f004:**
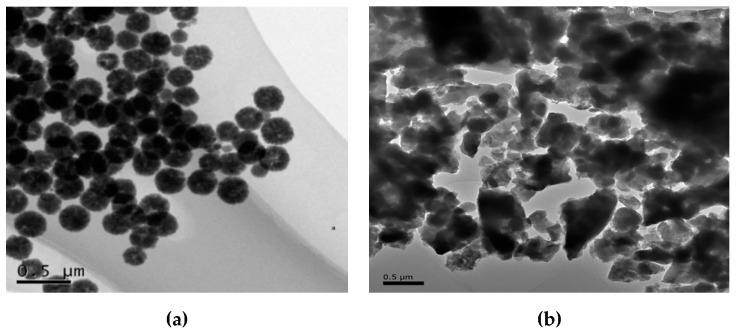
TEM of adsorbents, (**a**) uncoated Fe_3_O_4_ and (**b**) Fe_3_O_4_@mZrO_2_-Re (Re = Y/La/Ce).

**Figure 5 nanomaterials-11-02177-f005:**
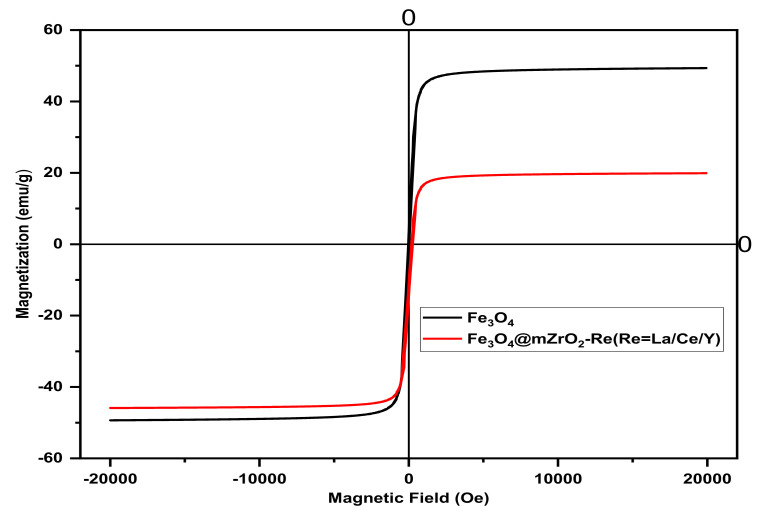
VSM Spectra of Fe_3_O_4_ and Fe_3_O_4_@mZrO_2_-Re (Re = Y/La/Ce).

**Figure 6 nanomaterials-11-02177-f006:**
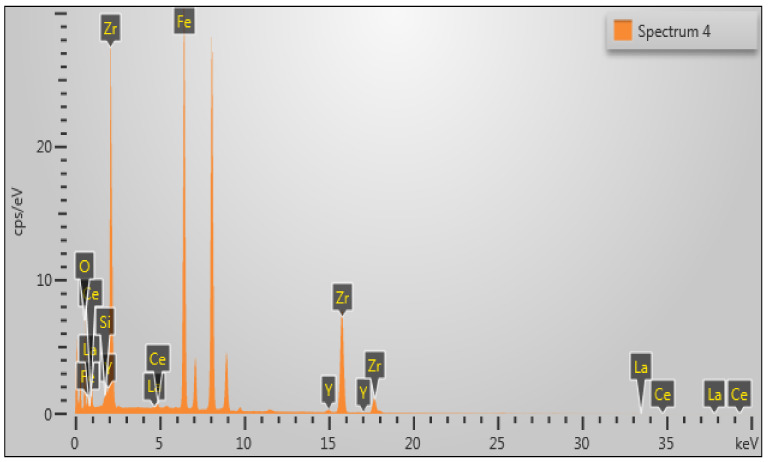
Energy Dispersive spectroscopy (EDS) patterns for Fe_3_O_4_@mZrO_2_-Re (Re = Y/La/Ce).

**Figure 7 nanomaterials-11-02177-f007:**
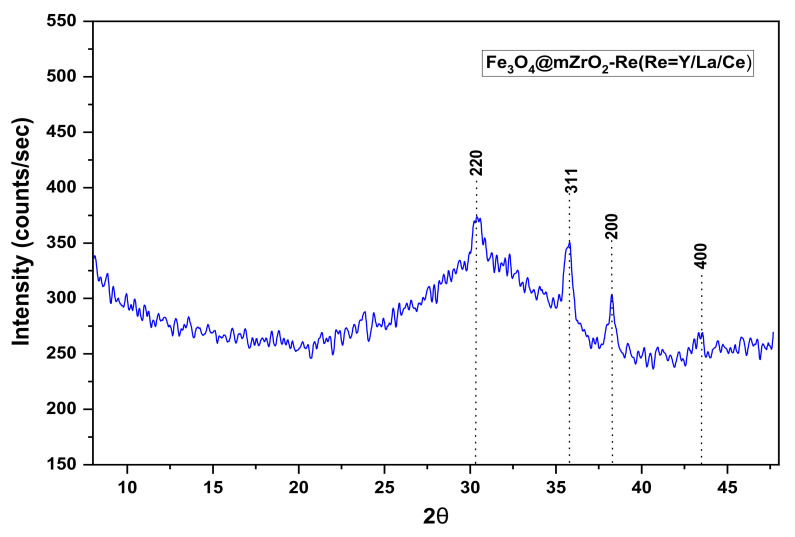
XRD spectrum of Fe_3_O_4_@mZrO_2_-Re (Re = Y/La/Ce).

**Figure 8 nanomaterials-11-02177-f008:**
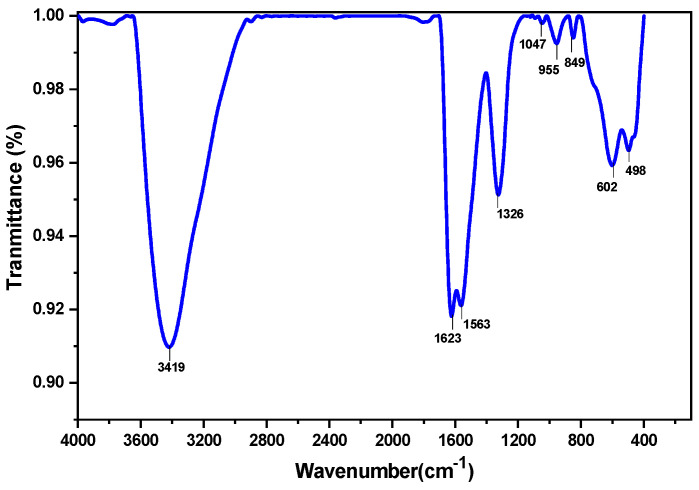
FTIR spectrum of Fe_3_O_4_@mZrO_2_-Re (Re = Y/La/Ce).

**Figure 9 nanomaterials-11-02177-f009:**
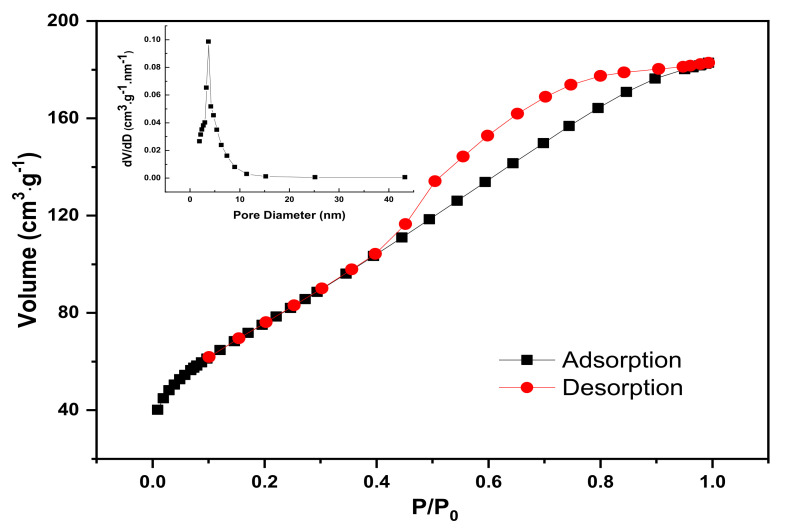
N_2_ adsorption-desorption Isotherm and Pore size distribution curve of Fe_3_O_4_@mZrO_2_-Re (Re = Y/La/Ce).

**Figure 10 nanomaterials-11-02177-f010:**
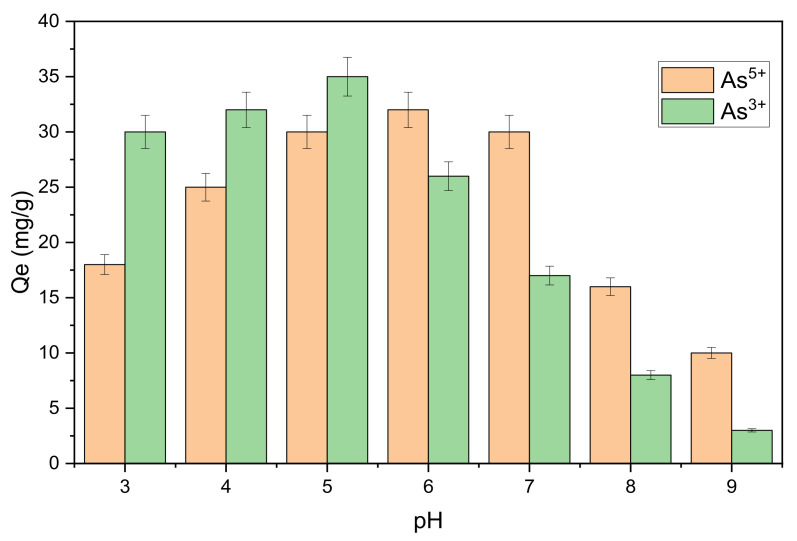
Effect of pH on As^3+^ and As^5+^ adsorption.

**Figure 11 nanomaterials-11-02177-f011:**
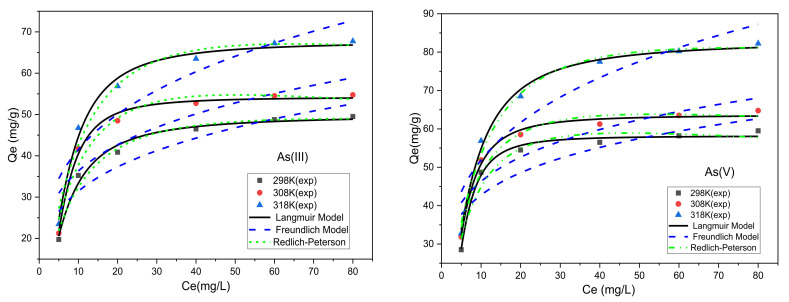
Fitting of Fe_3_O_4_@mZrO_2_-Re (Re = Y/La/Ce) for adsorption isotherm of As^3+^ and As^5+^.

**Figure 12 nanomaterials-11-02177-f012:**
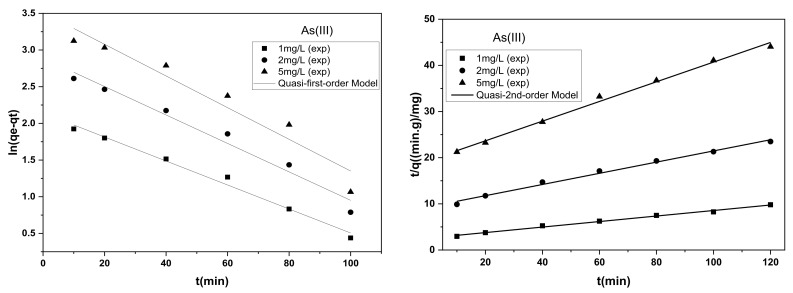
Quasi-first-order kinetics and quasi-second-order adsorption kinetic curves of As(III) and As(V).

**Figure 13 nanomaterials-11-02177-f013:**
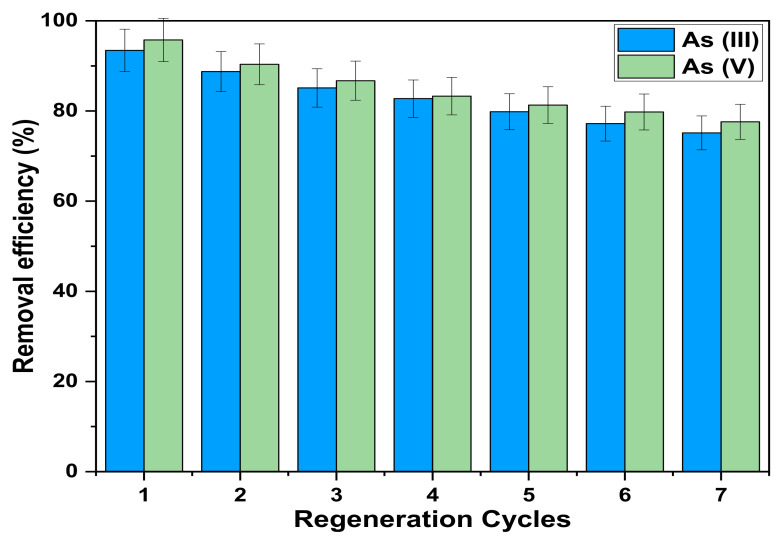
Regeneration cycles of As(III), As(V).

**Table 1 nanomaterials-11-02177-t001:** Table of uniform design U_14_*(14^5^).

	Independent Variable	I	II	III	IV	V
Experiment Seq.						
1st		1	4	7	11	13
2nd		2	8	14	7	11
3rd		3	12	6	3	9
4th		4	1	13	14	7
5th		5	5	5	10	5
6th		6	9	12	6	3
7th		7	13	4	2	1
8th		8	2	11	13	14
9th		9	6	3	9	12
10th		10	10	10	5	10
11th		11	14	2	1	8
12th		12	3	9	12	6
13th		13	7	1	8	4
14th		14	11	8	4	2

**Table 2 nanomaterials-11-02177-t002:** Guidelines for selecting columns of generating vectors in U_14_*(14^5^).

Number of Variables	Column Number	Deviation
2	I, IV	0.0957
3	I, II, III	0.1455
4	I, II, III, V	0.2091

**Table 3 nanomaterials-11-02177-t003:** Dose for selected parameters.

Variable	Level						
	1	2	3	4	5	6	7
Fe_3_O_4_ Dosage (X_1_)/mol	0.0004	0.00045	0.0005	0.00055	0.0006	0.00065	0.0007
TMAOH Dosage(X_2_)/mol	0.004	0.0045	0.005	0.0055	0.006	0.0065	0.007
Zirconium-Butoxide(X_3_)/mol	0.003	0.0035	0.004	0.0045	0.005	0.0055	0.006

**Table 4 nanomaterials-11-02177-t004:** Arrangements of uniform design and test results.

Experiment Sequence	Code			Actual Value			Experimental Values	Predicted Values
	X_1_	X_2_	X_3_	Fe_3_O_4_ (mol) (X_1_)	TMAOH (mol) (X_2_)	Zirconium(IV) Butoxide (mol) (X_3_)	Y (mmol/g)	Y′ (mmol/g)
1st	1	4	7	0.0004	0.0045	0.0045	0.6322	0.6122
2nd	2	8	14	0.0004	0.0055	0.006	0.6643	0.6443
3rd	3	12	6	0.00045	0.0065	0.004	0.5224	0.5324
4th	4	1	13	0.00045	0.004	0.006	0.5356	0.5556
5th	5	5	5	0.0005	0.005	0.004	0.6636	0.6436
6th	6	9	12	0.0005	0.006	0.0055	0.6567	0.6767
7th	7	13	4	0.00055	0.007	0.0035	0.4643	0.4543
8th	8	2	11	0.00055	0.004	0.0055	0.6743	0.6943
9th	9	6	3	0.0006	0.005	0.0035	0.5928	0.5728
10th	10	10	10	0.0006	0.006	0.005	0.6475	0.6875
11th	11	14	2	0.00065	0.007	0.003	0.4553	0.4353
12th	12	3	9	0.00065	0.0045	0.005	0.6274	0.6174
13th	13	7	1	0.0007	0.0055	0.003	0.5447	0.5247
14th	14	11	8	0.0007	0.0065	0.0045	0.5783	0.5913

**Table 5 nanomaterials-11-02177-t005:** Orthogonal Design arrangement of factors.

Number	Factors				Results
	A (Y)	B (La)	C (Ce)	D (Empty List)	(mg/g)
1	1 (0.5%)	1 (0.2%)	1 (0.5%)	1	29.52
2	1	2 (0.4%)	2 (0.8%)	2	30.05
3	1	3 (0.8%)	3 (1.2%)	3	29.16
4	2 (1%)	1	2	3	32.15
5	2	2	3	1	31.32
6	2	3	1	2	30.89
7	3 (2%)	1	3	2	29.16
8	3	2	1	3	27.65
9	3	3	2	1	28.54
Excellent level	A2	B1	C2		
Excellent combination	A2B1C2				

**Table 6 nanomaterials-11-02177-t006:** Elemental composition of Fe_3_O_4_@mZrO_2_-Re (Re = Y/La/Ce).

S/No	Element	Before Doping	After Doped
		Weight %	Weight %
1	O	38.16	37.2
2	Fe	9.08	12.1
3	Zr	52.76	48.1
4	Ce	0.1<	1.02
5	Y	0.1<	1.08
6	La	0.1<	0.45
7	Si	0.1<	0.1<
		100	99.95

**Table 7 nanomaterials-11-02177-t007:** Adsorption Isothermal data of Fe_3_O_4_@mZrO_2_-Re (Re = Y/La/Ce) for As^3+^ and As^5+^.

Model	Parameters	As(III)			As(V)		
		Temperature (K)					
		298	308	318	298	308	318
Langmuir	q_m_ (mg/g)	50.17	54.31	68.33	59.5	63.67	84.23
	K_L_ (L/mg)	0.102	0.048	0.067	0.136	0.095	0.104
	R_L_	0.662	0.806	0.748	0.594	0.677	0.556
	R^2^	0.987	0.992	0.991	0.991	0.991	0.994
Freundlich	K _F_	17.42	20.99	18.74	25.4	28.92	29.45
	n	3.97	4.23	3.22	4.87	5.34	3.92
	R^2^	0.864	0.781	0.841	0.753	0.793	0.87
Redlich–Peterson	K	6.86	6.69	7.36	10.09	10.96	10.4
	α	0.101	0.053	1.12	0.113	0.099	0.081
	β	1.05	1.16	1.12	1.077	1.108	1.077
	R^2^	0.978	0.962	0.974	0.953	0.966	0.988

**Table 8 nanomaterials-11-02177-t008:** Comparison of adsorption capacity of Fe_3_O_4_@mZrO_2_-Re with other relevant adsorbents.

S#	Adsorbent	Adsorption Capacity mg/g		pH	Reference
		As(III)	As(V)		
1	Fe_3_O_4_@mZrO_2_-Re (Re = Y/La/Ce)	68.33	84.23	5–6	This study
2	CeO_2_-ZrO_2_	9.2	27.5	6.7–7.1	[31]
3	Mesoporous Zirconia nanostructures (MZN)	105.03	110.29	5.0–9.0	[29]
4	Magnetic Fe_3_O_4_@Carbon Encapsulates	17.90	24.40	7.0	[37]
5	r-Fe_2_O_3_@ZrO_2_	62.2	18.3	9.0	[38]
6	Amorphous ZrO_2_ Nanomaterials	83	32.4	7.0	[39]
7	NZVI-RGO nanoparticles	35.8	29.04	7.0	[40]
8	Ce-Al Nanostructured mixed oxide	n.a	21	5.0	[22]
9	Zirconium based nanoparticles	n.a	35	8.0	[41]
10	Ce(III) doped Titanium-Oxide	55.3	44.9	6.0	[42]
11	CeO_2_ modified Activated carbon	36.77	43.60	5.0	[43]
12	Ce modified chitosan ultrafine nanobiosorbent	57.5	n.a	8	[44]

**Table 9 nanomaterials-11-02177-t009:** Quasi-first-order and quasi-second-order adsorption kinetic fitting data for As(III) and As(V).

Metal	Initial Concentration (mg/L)	Quasi First Order Dynamics Model			Quasi Second-Order Dynamics Model		
		R^2^	*q_e_* (cal) (mg/g)	K_1_ min^−1^	R^2^	*q_e_* (cal) (mg/g)	K_2_ g mg^−1^ min^−1^
As^3+^	1	0.987	9.8	0.006	0.996	9.8	0.0013
	2	0.972	23.5	0.018	0.995	23.5	0.0015
	5	0.934	43.5	0.014	0.997	44	0.0023
As^5+^	1	0.921	9.5	0.021	0.993	11.5	0.0015
	5	0.954	46.5	0.022	0.991	50.75	0.0030

## Data Availability

The data is available on reasonable request from the corresponding author or first author.

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
