# Peer review of "Synthesis of Fe3O4@mZrO2-Re (Re = Y/La/Ce) by Using Uniform Design, Surface Response Methodology, and Orthogonal Design & Its Application for As3+ and As5+ Removal"

_nanomaterials, 2021, doi:10.3390/nano11092177_

Round 1
Reviewer 1 Report
Removal of arsenic compounds are very important, and this work is very important. The experiments seem conducted well, but some of the discussion and explanation are inappropriate or insufficient. Detailed comments are indicated below.
Experimental section.
The conditions for the synthesis of magnetite should be indicated briefly for readers’ convenience (e.g., amount, temperature, time).
The X-ray source for XRD measurements must be indicated.
The concentrations of As must be indicated if the calculation of Y values uses any experiments.
The design and analytical methods (e.g., line 120-123) should be explained in the introduction or results and discussion sections. The experimental section should simply describe only the procedure with the software used.
Variables and constants (beta, X, and epsilon) in Scheme 1 are not described, and I could not understand how the Y values are obtained. Accordingly, the following comments might contain inadequate ones due to my confusion.
Result and discussion
Table 1-3 and Figure S1 in SI should be moved to the main manuscript, or the table and figure numbers in the main text should be renumbered from.
Table 4. The units for the code X1-3 should be indicated. For the response value indicating the capacity, relative molar amount toward the weight of the adsorbent (e.g., mol/g) is more understandable (I presume /g is missing). Are the response values indicated in Table 4 Y, not Y’?
I think that the molar amounts for the actual values in mmol, micro-mole, or mmol/mg are better.
Figure (S)1 and text. Y and Y’ should be clearly defined at the first appearance (both in the text and figure). Y’ seems not defined.
Figure 2. “Magnetic core” in the bottom plain should be unified with the parameter name used in this work (Fe3O4) to avoid confusion. I think “Factor 1, Fe3O4” and “Factor 3, Zr(OBu)4” is more understandable.
The results for only As(V) are indicated before line 230, but the authors expand the best combination optimized for As(V) to As(III). The adsorbent optimized for As(V) may be used for As(III), but it is plausibly not the best for As(III).
Figure 4. Black color in TEM images does not reflect the optical color of materials (mostly scattering contrast in this case).
Figure 5 and Table 6. The signals for the rare earth elements are almost invisible. It seems that the intensities of the signals are in the noise level (below threshold of detection). The expanded spectrum and the spectrum of non-doped one should be indicated. At least, the significant figures of the percentages are inappropriately more.
Figure 6. The intensity of the diffraction peaks is very weak for Fe3O4. Does this originate from the coating damaging the crystallinity, smaller crystallite size, or the degradation of the crystalline structure during modification? The spectrum of bare magnetite should also be indicated for discussion.
Figure 7. I could not be convinced to the properness of the spectrum indicating the clear signals toward the very low transmittance and narrow transmittance range (< 1%). I presume that the vertical axis is not in percent or inappropriate smoothing was given.
If the signal for C-H is included, the source should be discussed. If ethylene glycol is included, the spectrum should be indicated.
Line 278. Is the hydroxide (OH-) originates from TMAOH? Any counter cation is necessary. In addition, this absorption may also be assignable to water or alcohols adsorbed, or FeO-H and ZrO-H bonds. Any clearer proof should be indicated.
The comparison of the bare and coated adsorbents will give more information.
Adsorption isotherms. The font style for italic is not appropriate and not unified.
The ability of this adsorbent should be compared with reported materials.
Author Response
Comments and Suggestions for Author:
Removal of arsenic compounds are very important, and this work is very important. The experiments seem conducted well, but some of the discussion and explanation are inappropriate or insufficient. Detailed comments are indicated below.
Answer: Thank you very much for your suggestions and comments. We tried to address all your raised points, and made modifications or improvements accordingly. Some new references added against some revisions. Due to being graduated from the school, it’s difficult to conduct any new test, and also schools are closed due to recent pandemic outbreak in our province
Experimental section.
Comment 1: The conditions for the synthesis of magnetite should be indicated briefly for readers’ convenience (e.g., amount, temperature, time).
Answer: Thank you very much for pointing out this, the mentioned Fe3O4 procedure has been added in the material synthesis section (2.3).
Comment 2: The X-ray source for XRD measurements must be indicated.
Answer: The X-Ray source for this study is added to characterization methods (2.2).
Comment 3: The concentrations of As must be indicated if the calculation of Y values uses any experiments.
Answer: The amount of As5+ and other basic experimental conditions are added in section 3.1.1
Comment 4: The design and analytical methods (e.g., line 120-123) should be explained in the introduction or results and discussion sections. The experimental section should simply describe only the procedure with the software used.
Variables and constants (beta, X, and epsilon) in Scheme 1 are not described, and I could not understand how the Y values are obtained. Accordingly, the following comments might contain inadequate ones due to my confusion.
Answer: we have modified the portion as per your suggestion, because its already mentioned in the introduction section so the irrelevant lines are removed. All the mentioned parts are revisited and modified as per suggestions.
Result and discussion
Comment 5: Table 1-3 and Figure S1 in SI should be moved to the main manuscript, or the table and figure numbers in the main text should be renumbered from.
Answer: By agreeing with your suggestion, all the mentioned tables (1-3) and figure-1 are being added to the main body, and removed from the annexures.
Comment 6: Table 4. The units for the code X1-3 should be indicated. For the response value indicating the capacity, relative molar amount toward the weight of the adsorbent (e.g., mol/g) is more understandable (I presume /g is missing). Are the response values indicated in Table 4 Y, not Y’?
Answer. The relevant portion is modified according to your suggestions. Predicted values are added to the table 4 and also in the text some more relevant lines are added to express the details of Y, and Y
Comment 7: I think that the molar amounts for the actual values in mmol, micro-mole, or mmol/mg are better.
Answer: I think the values should be mmol, and literature also suggests, for reference we are attaching a reference here. https://doi.org/10.1007/s11356-)
Comment 8: Figure (S)1 and text. Y and Y’ should be clearly defined at the first appearance (both in the text and figure). Y’ seems not defined.
Answer: Thank you very much for mentioning this, and a relevant detail is already added tin the text and also a column is inserted in table 4 for predicted values.
Comment 9: Figure 2. “Magnetic core” in the bottom plain should be unified with the parameter name used in this work (Fe3O4) to avoid confusion. I think “Factor 1, Fe3O4” and “Factor 3, Zr(OBu)4” is more understandable.
Answer: Your suggested points are noted, and modified accordingly in the Figure 2 and Figure 3, and also in the relevant text.
Comment 10: The results for only As(V) are indicated before line 230, but the authors expand the best combination optimized for As(V) to As(III). The adsorbent optimized for As(V) may be used for As(III), but it is plausibly not the best for As(III).
Answer: Thank you for pointing our this, the corrections have already made and already removed the mentioned parts, and modified accordingly.
Comment 11: Figure 4. Black color in TEM images does not reflect the optical color of materials (mostly scattering contrast in this case).
Answer: These are pictures received after TEM, and selected best pictures. This portion is modified and removed irrelevant lines, and consisted on scattered.
Comment 12: Figure 5 and Table 6. The signals for the rare earth elements are almost invisible. It seems that the intensities of the signals are in the noise level (below threshold of detection). The expanded spectrum and the spectrum of non-doped one should be indicated. At least, the significant figures of the percentages are inappropriately more.
Answer: A clear EDS Spectrum picture is added to the mentioned parts. The possible reason may be the amount of the mentioned elements were in low % so may be the peaks are not that much strong.
Comment 13: Figure 6. The intensity of the diffraction peaks is very weak for Fe3O4. Does this originate from the coating damaging the crystallinity, smaller crystallite size, or the degradation of the crystalline structure during modification? The spectrum of bare magnetite should also be indicated for discussion.
Answer: Modified the XRD pictures with bit clear peaks. As the magnetite was modified with surface coating and doping therefore only focused was made on the adsorbent.
Comment 14: Figure 7. I could not be convinced to the properness of the spectrum indicating the clear signals toward the very low transmittance and narrow transmittance range (< 1%). I presume that the vertical axis is not in percent or inappropriate smoothing was given.
If the signal for C-H is included, the source should be discussed. If ethylene glycol is included, the spectrum should be indicated.
Asnwer: Ethylene glycol was only used for the synthesis of magnetic core (Fe3O4), while in the preparation of Fe3O4@mZrO2-Re (Re=Y/La/Ce) adsorbent EG was not used. The mentioned part is modified and corrected the mistake made.
Comment 15: Line 278. Is the hydroxide (OH-) originates from TMAOH? Any counter cation is necessary. In addition, this absorption may also be assignable to water or alcohols adsorbed, or FeO-H and ZrO-H bonds. Any clearer proof should be indicated. The comparison of the bare and coated adsorbents will give more information.
Answer: TMAOH is also used in the synthesis of materials, the related lines are updated accordingly.
Comment 16: Adsorption isotherms. The font style for italic is not appropriate and not unified.
Answer: This portion was revisited and modified according to your mentioned points.
Comment 17: The ability of this adsorbent should be compared with reported materials.
Answer: A comparative table (table 8) is included to the manuscript to compare the capacity with the materials.
Reviewer 2 Report
Recommendation: Major revisions
Comments:
In this work, magnetic Fe3O4 used as core, coated with ZrO2, and triple doped with three rare earth elements((Y/La/Ce), and a multi-staged core-shelled rare earth doped zirconia adsorbent was prepared by applying uniform design U14, Response Surface methodology, and orthogonal design, to remove the As3+ and As5+ from the aqueous solution. The topic is very interesting, the writing is clear, and the manuscript does not contain technical errors. It is impressive and suit well to the scope of Nanomaterials. The manuscript can be published with Major revisions.
- How about the chemical stability of magnetic core under different pH?how about The leakage of Fe3+?
- From the Fig.9, the initial pH value of the solution has a substantial effect on the adsorption, more explanation should be provided for understanding the different trend for As3+ and As5+. The Zeta potential of adsorbent is necessary to know about the binding mechanism, so it is needed to do it.
- Unite of the calculated constants from model should be carefully checked.
- Saturation magnetic intensity of as-prepared adsorbents should be tested.
- How about the specific adsorption for As3+ and As5+? why?
- The effect of coexisting ions to the uptake performance should be considered.
- The error bar for the results from batch mode experiments should be supplied.
- The comparison with the previously reported adsorbents should be summarized, especially the important parameters.
- Some minor format errors in the text and references need to be carefully revised.
Author Response
Comments and Suggestions for Authors
Recommendation: Major revisions
Comments:
In this work, magnetic Fe3O4 used as core, coated with ZrO2, and triple doped with three rare earth elements((Y/La/Ce), and a multi-staged core-shelled rare earth doped zirconia adsorbent was prepared by applying uniform design U14, Response Surface methodology, and orthogonal design, to remove the As3+ and As5+ from the aqueous solution. The topic is very interesting, the writing is clear, and the manuscript does not contain technical errors. It is impressive and suit well to the scope of Nanomaterials. The manuscript can be published with Major revisions.
Reply: Thank you very much for your comments and suggestions regarding our manuscript. We tried our best to revise your mentioned parts, and finished the revision accordingly. In the previous manuscript by considering it lengthy we skipped some parts which you rightly point out. Here are then answers to your comments. Due to being graduated from the school, it’s difficult to conduct any new test, and also schools are closed due to recent pandemic outbreak in our province.
Comment 1: How about the chemical stability of magnetic core under different pH?how about The leakage of Fe3+?
Answer: We didn’t focus on this study by considering the fact the Fe3O4 is coated with Zirconia and doped with three rare earth metals so the magnetic core should be least affected or not considerable affection.
Comment 2: From the Fig.9, the initial pH value of the solution has a substantial effect on the adsorption, more explanation should be provided for understanding the different trend for As3+ and As5+. The Zeta potential of adsorbent is necessary to know about the binding mechanism, so it is needed to do it.
Answer: Thank you very for pointing out this, and by agreeing with your suggestion a detailed paragraph is added along with reasons with references to handle the mentioned issues. And I hope you will be satisfied with the answer.
Comment 3: Unite of the calculated constants from model should be carefully checked.
Answer: We again thoroughly checked the mentioned areas and corrections made in the adsorption isothermal parameters table and modified accordingly.
Comment 4: Saturation magnetic intensity of as-prepared adsorbents should be tested.
Answer: Thank you very much for pointing out this. We thought manuscript is very lengthy so parts of the study were not included, but now a VSM study (3.2.2) is added along with the Figure.
Comment 5: How about the specific adsorption for As3+ and As5+? why?
Comment 6: The effect of coexisting ions to the uptake performance should be considered.
Answer: Due to limited available time duration, closure of schools here we didn’t do the coexisting ions tests, and also some relevant studies also supports our research.
Comment 7: The error bar for the results from batch mode experiments should be supplied.
Answer: The error bars are added in pH and regeneration graphs.
Comment 8: The comparison with the previously reported adsorbents should be summarized, especially the important parameters.
Answer: By agreeing with your suggestion a Table (8) is added to the manuscript to compare this material with some relevant studies.
Comment 9: Some minor format errors in the text and references need to be carefully revised.
Answer: Manuscript is checked again twice for the mentioned errors and mistakes, and tried our best to remove the those, we are hopeful that this time the mentioned issues are well addressed.
Reviewer 3 Report
The manuscript deals with the synthesis and characterization of a complex oxide material with core-shell architecture, based on a Fe3O4 and a ZrO2 coating, to which rare-earth (Y, La and Ce) doping is added.
However, the manuscript proves serious shortages from the scientific and language viewpoints.
If the erroneous use of terms in English, typos and wrong English phrase formulations might be corrected or improved, the scientific evidence, message and language are not at a satisfactory level.
The manuscript contains many details regarding the technological steps and procedures. However, the experimental evidence does not support the authors’ description of the synthesized core-shell materials. For instance, no evidence whatsoever is presented by the TEM images regarding the claimed core-shell architecture, crystalline structure, local chemical composition, spatial distribution of the detected elements, etc. The EDS results are irrelevant in the absence of any necessary detail regarding the analyzed areas on the samples, probing mode, surface morphology of the samples. The XRD pattern has a low quality and the interpretation is, also, rather poor.
In conclusion, I regretfully consider that this manuscript is not adequate for Nanomaterials and should not be accepted for publication.
Author Response
Comments and Suggestions for Authors
The manuscript deals with the synthesis and characterization of a complex oxide material with core-shell architecture, based on a Fe3O4 and a ZrO2 coating, to which rare-earth (Y, La and Ce) doping is added.
However, the manuscript proves serious shortages from the scientific and language viewpoints.
If the erroneous use of terms in English, typos and wrong English phrase formulations might be corrected or improved, the scientific evidence, message and language are not at a satisfactory level.
The manuscript contains many details regarding the technological steps and procedures. However, the experimental evidence does not support the authors’ description of the synthesized core-shell materials. For instance, no evidence whatsoever is presented by the TEM images regarding the claimed core-shell architecture, crystalline structure, local chemical composition, spatial distribution of the detected elements, etc. The EDS results are irrelevant in the absence of any necessary detail regarding the analyzed areas on the samples, probing mode, surface morphology of the samples. The XRD pattern has a low quality and the interpretation is, also, rather poor.
In conclusion, I regretfully consider that this manuscript is not adequate for Nanomaterials and should not be accepted for publication.
Answer: Thank you very much for your time and input. We are sorry that we couldn’t meet your expectations, and tried our best to improve the portions as you identified. The mentioned typos, and other English phrase formulations are improved. We tried to present our manuscript with some references and we believe shortcomings have been improved.
Round 2
Reviewer 1 Report
The descriptions are improved, and I regard that this manuscript may be published after sufficient editing on styles and grammar.
Comment 8: Figure (S)1 and text. Y and Y’ should be clearly defined at the first appearance (both in the text and figure). Y’ seems not defined.
Answer: Thank you very much for mentioning this, and a relevant detail is already added tin the text and also a column is inserted in table 4 for predicted values.
Comment: Definitions should be indicated both in the text and tables in typical journals.
Comment 11: Figure 4. Black color in TEM images does not reflect the optical color of materials (mostly scattering contrast in this case).
Answer: These are pictures received after TEM, and selected best pictures. This portion is modified and removed irrelevant lines, and consisted on scattered.
Comment: I meant that black or dark in TEM are not the color of materials. The description in Line 270- should be as follows. In addition, amorphous nature cannot be judged from these images. High resolution images with lattice fringes or SAED are necessary.
Figure 4 shows the TEM images of Fe3O4 (a), and Fe3O4@mZrO2-Re (Re=Y/La/Ce) (b). Uncoated Fe3O4 was observed as dispersed spherical particles with diameters ranging X-Y um. The good dispersibility probably originates from TMAOH that effectively prevents from the agglomeration [22]. By contrast, Fe3O4@mZrO2-Re (Re=Y/La/Ce) exhibited uneven and larger morphologies probably due to the coating with ZrO2 doped with the rare earth elements.
Comment 12: Figure 5 and Table 6. The signals for the rare earth elements are almost invisible. It seems that the intensities of the signals are in the noise level (below threshold of detection). The expanded spectrum and the spectrum of non-doped one should be indicated. At least, the significant figures of the percentages are inappropriately more.
Answer: A clear EDS Spectrum picture is added to the mentioned parts. The possible reason may be the amount of the mentioned elements were in low % so may be the peaks are not that much strong.
Comment: The two spectra in Figure 6 seem identical. What do these mean? The legend and the spectra should be corrected. As I commented in the first review, EDS inherently do not have accuracy in the indicated significant figures. The values below 1% are not reliable. The numbers in Table 6 should be, for example, as follows.
38.2 37.2
9.1 12.1
52.8 48.1
0.1 > 1.0
0.1 > 1.1
0.1 > 0.5
0.1 > 0.1 >
Comment 13: Figure 6. The intensity of the diffraction peaks is very weak for Fe3O4. Does this originate from the coating damaging the crystallinity, smaller crystallite size, or the degradation of the crystalline structure during modification? The spectrum of bare magnetite should also be indicated for discussion.
Answer: Modified the XRD pictures with bit clear peaks. As the magnetite was modified with surface coating and doping therefore only focused was made on the adsorbent.
Comment: The XRD profile with weak signals toward the halo peak probably indicate that the magnetite contains both crystalline and amorphous parts. The XRD profile of the magnetite before modification is informative to clarify the reason. Even if the XRD profile of the original magnetite cannot be added, the presence of the amorphous part should be mentioned. In addition, the sentence in Line 320 is inappropriate. The lattice structure is face-centered cubic, but the particles are not cubical. The following explanation is better.
The diffraction peaks at 2theta = 30.4, 35.8, 38.2, and 43Ëš are assignable to the 220, 311, 200, and 400 lattice planes of face-centered cubic magnetite (JCPDS 19-0629), respectively.
Comment 14: Figure 7. I could not be convinced to the properness of the spectrum indicating the clear signals toward the very low transmittance and narrow transmittance range (< 1%). I presume that the vertical axis is not in percent or inappropriate smoothing was given.
If the signal for C-H is included, the source should be discussed. If ethylene glycol is included, the spectrum should be indicated.
Asnwer: Ethylene glycol was only used for the synthesis of magnetic core (Fe3O4), while in the preparation of Fe3O4@mZrO2-Re (Re=Y/La/Ce) adsorbent EG was not used. The mentioned part is modified and corrected the mistake made.
Comment: As I commented, the vertical resolution of this spectrum is unbelievably high toward the very low transmittance. The measurement method (ATR? KBr disk?) with the measurement conditions should be added. The accuracy of the wavenumber in Line 91 is also unbelievable. Typical IR spectroscopic measurements are conducted with the step of 1-4 cm-1. These features are not attainable with typical IR instruments. Peak assignments and positions are agreeable, but the indicated detail should be reconfirmed.
Author Response
Comments and Suggestions for Authors
The descriptions are improved, and I regard that this manuscript may be published after sufficient editing on styles and grammar.
Answer: Thank you very much again for the feedback and kind suggestions, which are very helpful to improve our manuscript. Your all points are very important and valid, and we tried our best to revise it accordingly. We believe that manuscript is now up to your expectations and we will be waiting for your further feedback.
Comment 8: Figure (S)1 and text. Y and Y’ should be clearly defined at the first appearance (both in the text and figure). Y’ seems not defined.
Answer: Thank you very much for mentioning this, and a relevant detail is already added tin the text and also a column is inserted in table 4 for predicted values.
Comment: Definitions should be indicated both in the text and tables in typical journals.
New Answer: The definitions of experimental values (Y) and predicted values (Y’) are briefly added in section 3.1, in text, table 4, and in figure 1 too.
Comment 11: Figure 4. Black color in TEM images does not reflect the optical color of materials (mostly scattering contrast in this case).
Answer: These are pictures received after TEM, and selected best pictures. This portion is modified and removed irrelevant lines, and consisted on scattered.
Comment: I meant that black or dark in TEM are not the color of materials. The description in Line 270- should be as follows. In addition, amorphous nature cannot be judged from these images. High resolution images with lattice fringes or SAED are necessary.
Figure 4 shows the TEM images of Fe3O4 (a), and Fe3O4@mZrO2-Re (Re=Y/La/Ce) (b). Uncoated Fe3O4 was observed as dispersed spherical particles with diameters ranging X-Y um. The good dispersibility probably originates from TMAOH that effectively prevents from the agglomeration [22]. By contrast, Fe3O4@mZrO2-Re (Re=Y/La/Ce) exhibited uneven and larger morphologies probably due to the coating with ZrO2 doped with the rare earth elements.
New Answer: Thank you very much for the corrections, the mentioned TEM parts are modified according. Your suggestions are more appropriate and by agreeing that we added in the manuscript accordingly.
Comment 12: Figure 5 and Table 6. The signals for the rare earth elements are almost invisible. It seems that the intensities of the signals are in the noise level (below threshold of detection). The expanded spectrum and the spectrum of non-doped one should be indicated. At least, the significant figures of the percentages are inappropriately more.
Answer: A clear EDS Spectrum picture is added to the mentioned parts. The possible reason may be the amount of the mentioned elements were in low % so may be the peaks are not that much strong.
Comment: The two spectra in Figure 6 seem identical. What do these mean? The legend and the spectra should be corrected. As I commented in the first review, EDS inherently do not have accuracy in the indicated significant figures. The values below 1% are not reliable. The numbers in Table 6 should be, for example, as follows.
38.2 37.2
9.1 12.1
52.8 48.1
0.1 > 1.0
0.1 > 1.1
0.1 > 0.5
0.1 > 0.1 >
New Answer: The table is modified accordingly but while thinking of your comment I added less than (<) symbol while you mentioned greater than (>), I this ok? There are many papers reported some elements as mentioned as 0.00 in the EDS tables, so I added this as it received from the machine.
Comment 13: Figure 6. The intensity of the diffraction peaks is very weak for Fe3O4. Does this originate from the coating damaging the crystallinity, smaller crystallite size, or the degradation of the crystalline structure during modification? The spectrum of bare magnetite should also be indicated for discussion.
Answer: Modified the XRD pictures with bit clear peaks. As the magnetite was modified with surface coating and doping therefore only focused was made on the adsorbent.
Comment: The XRD profile with weak signals toward the halo peak probably indicate that the magnetite contains both crystalline and amorphous parts. The XRD profile of the magnetite before modification is informative to clarify the reason. Even if the XRD profile of the original magnetite cannot be added, the presence of the amorphous part should be mentioned. In addition, the sentence in Line 320 is inappropriate. The lattice structure is face-centered cubic, but the particles are not cubical. The following explanation is better.
The diffraction peaks at 2theta = 30.4, 35.8, 38.2, and 43Ëš are assignable to the 220, 311, 200, and 400 lattice planes of face-centered cubic magnetite (JCPDS 19-0629), respectively.
New Answer: Thank you very for the feedback, the XRD characterization in the section 3.2.4 is modified as per your recommendations and a reference has been added too.
Comment 14: Figure 7. I could not be convinced to the properness of the spectrum indicating the clear signals toward the very low transmittance and narrow transmittance range (< 1%). I presume that the vertical axis is not in percent or inappropriate smoothing was given.
If the signal for C-H is included, the source should be discussed. If ethylene glycol is included, the spectrum should be indicated.
Answer: Ethylene glycol was only used for the synthesis of magnetic core (Fe3O4), while in the preparation of Fe3O4@mZrO2-Re (Re=Y/La/Ce) adsorbent EG was not used. The mentioned part is modified and corrected the mistake made.
Comment: As I commented, the vertical resolution of this spectrum is unbelievably high toward the very low transmittance. The measurement method (ATR? KBr disk?) with the measurement conditions should be added. The accuracy of the wavenumber in Line 91 is also unbelievable. Typical IR spectroscopic measurements are conducted with the step of 1-4 cm-1. These features are not attainable with typical IR instruments. Peak assignments and positions are agreeable, but the indicated detail should be reconfirmed.
Answer new: Thank you very much for pointing out this, we didn’t include the details about characterization means of FTIR before by considering manuscript as lengthy, but now added details in the section 2.2 as per your identification.
Reviewer 2 Report
The quality of this revised manuscript has been improved and it can be published in the journal of Nanomaterials.
Author Response
Comments and Suggestions for Authors
The quality of this revised manuscript has been improved and it can be published in the journal of Nanomaterials.
Answer: Thank you very much for your kind feedback and suggestions, which are very helpful in improving our manuscript.
Reviewer 3 Report
Reading the revised manuscript, I have noticed the text modifications operated by the authors. However, the manuscript does not contain relevant experimental evidence in order to clearly prove the claimed core-shell architecture, which should have been demonstrated by TEM. The two TEM images offer no information regarding the crystalline structure, local chemical composition, spatial distribution of the detected elements, or any relation between the two samples to prove the ZrO2 coating around the Fe3O4 core, not to mention the rare-earth doping of the shell. The EDS results are irrelevant in the absence of any necessary detail regarding the analyzed areas on the samples, probing mode, surface morphology of the samples. The XRD pattern has a low quality and the interpretation is rather poor.
Despite the text modifications (some useful, some not so inspired), there is no improvement concerning scientific content, including experimental data and interpretation. Consequently, I regretfully maintain my position that, in my opinion, the manuscript does not meet the necessary scientific level for publication in Nanomaterials.
Author Response
Comments and Suggestions for Authors
Reading the revised manuscript, I have noticed the text modifications operated by the authors. However, the manuscript does not contain relevant experimental evidence in order to clearly prove the claimed core-shell architecture, which should have been demonstrated by TEM. The two TEM images offer no information regarding the crystalline structure, local chemical composition, spatial distribution of the detected elements, or any relation between the two samples to prove the ZrO2 coating around the Fe3O4 core, not to mention the rare-earth doping of the shell. The EDS results are irrelevant in the absence of any necessary detail regarding the analyzed areas on the samples, probing mode, surface morphology of the samples. The XRD pattern has a low quality and the interpretation is rather poor.
Despite the text modifications (some useful, some not so inspired), there is no improvement concerning scientific content, including experimental data and interpretation. Consequently, I regretfully maintain my position that, in my opinion, the manuscript does not meet the necessary scientific level for publication in Nanomaterials.
Answer: Thank you very much for your kind feedback, we appreciate your time and comments regarding our manuscript.